# Evaluating Phonon Characteristics by Varying the Layer and Interfacial Thickness in Novel Carbon-Based Strained-Layer Superlattices

**Devki N. Talwar** [1,2,*] and **Piotr Becla** [3]

[1] Department of Physics, University of North Florida, 1 UNF Drive, Jacksonville, FL 32224, USA
[2] Department of Physics, Indiana University of Pennsylvania, 975 Oakland Avenue, 56 Weyandt Hall, Indiana, PA 15705, USA
[3] Department of Materials Science and Engineering, Massachusetts Institute of Technology, Cambridge, MA 02139, USA; becla@mit.edu
[*] Correspondence: d.talwar@unf.edu; Tel.: +1-724-762-7719

**Abstract:** Systematic results of lattice dynamical calculations are reported as a function of m and n for the novel $(SiC)_m/(GeC)_n$ superlattices (SLs) by exploiting a modified linear-chain model and a realistic rigid-ion model (RIM). A bond polarizability method is employed to simulate the Raman intensity profiles (RIPs) for both the ideal and graded $(SiC)_{10-\Delta}/(Si_{0.5}Ge_{0.5}C)_\Delta/(GeC)_{10-\Delta}/(Si_{0.5}Ge_{0.5}C)_\Delta$ SLs. We have adopted a virtual-crystal approximation for describing the interfacial layer thickness, $\Delta$ ($\equiv 0$, 1, 2, and 3 monolayers (MLs)) by selecting equal proportions of SiC and GeC layers. Systematic variation of $\Delta$ has initiated considerable upward (downward) shifts of GeC-(SiC)-like Raman peaks in the optical phonon frequency regions. Our simulated results of RIPs in SiC/GeC SLs are agreed reasonably well with the recent analyses of Raman scattering data on graded short-period GaN/AlN SLs. Maximum changes in the calculated optical phonons (up to $\pm \sim 47$ cm$^{-1}$) with $\Delta = 3$, are proven effective for causing accidental degeneracies and instigating localization of atomic displacements at the transition regions of the SLs. Strong $\Delta$-dependent enhancement of Raman intensity features in SiC/GeC are considered valuable for validating the interfacial constituents in other technologically important heterostructures. By incorporating RIM, we have also studied the phonon dispersions $[\omega_j^{SL}(\vec{q})]$ of $(SiC)_m/(GeC)_n$ SLs along the growth [001] as well as in-plane [100], [110] directions [i.e., perpendicular to the growth]. In the acoustic mode regions, our results of $\omega_j^{SL}(\vec{q})$ have confirmed the formation of mini-gaps at the zone center and zone edges while providing strong evidences of the anti-crossing and phonon confinements. Besides examining the angular dependence of zone-center optical modes, the results of phonon folding, confinement, and anisotropic behavior in $(SiC)_m/(GeC)_n$ are compared and contrasted very well with the recent first-principles calculations of $(GaN)_m/(AlN)_n$ strained layer SLs.

**Keywords:** novel $(SiC)_m/(GeC)_n$ superlattices; interfacial layer thickness; Raman intensity profiles; lattice dynamics; localization of atomic displacements

## 1. Introduction

Semiconductor quantum wells (QWs) and superlattices (SLs) have formed the basis of fabricating many modern electronic and optoelectronic devices, including the light-emitting diodes (LEDs), laser diodes (LDs), field-effect transistors (FETs), etc. [1–10]. Compared with II-VI and III-V compounds, the epitaxial growth of C-based zinc-blende (zb) IV-IV (XC with X = Si, Ge, Sn) binary materials, alloys and heterostructures (i.e., QWs, SLs, etc.) of higher thermal conductivity, wider electronic energy bandgaps, and higher mechanical strength have recently stimulated interest among the technologists to design different types of device structures (e.g., meta-photonic heterostructures, holographic displays, lasers, etc.) and for the scientists to evaluate their basic traits [1–10]. The progress in device engineering has

demanded careful selection of the C-based wide-bandgap $E_g$ (SiC = 2.42 eV; GeC = 1.52 eV) materials which maintain physical properties both at elevated temperatures and higher radiation levels [2–9]. A major precondition for realizing commercial wide-bandgap hetero-junction-based devices requires large-area substrates for the preparation of functional multilayer structures.

Like GaN/Al(In)N and GaN/Al(Ga)N SLs, the SiC/Ge(Si)C based materials can be grown on SiC substrates. In the binary strained layer SiC/GeC heterostructure, one might expect diffusions of both the cation and/or anion atoms near the sharp interfaces. This distribution of atoms would suggest the formation of an ultrathin interfacial region with random occupations of Si, Ge, and/or C atoms [10–23]. In III-V-N-based QWs, the existence of nano-crystallite interfaces has been observed [21] which provided strong impacts on many of their optical and/or electro-optical characteristics. Despite some early efforts made in the GaN/InGaN/AlGaN structures by using Raman scattering spectroscopy (RSS) [10,21], the challenges still persist of monitoring and accurately estimating the inter-facial layer thickness in III-V-N, IV-IV QWs, SLs and many other technologically important heterostructures. In graded $(SiC)_m/(GeC)_n$ SLs (where m and n represent the numbers of SiC and GeC monolayers), the interfaces can act as potential scatters to play crucial roles in the thermal energy transport [10–23]. Theoretical lattice dynamical calculations by including the appropriate thickness of an interfacially layered structure has been and still is quite a challenging task. By incorporating atomic interfacial structures, our simulation of phonons in strained-layer SiC/GeC SLs can be considered as a major step forward for apprehending the heat transport in thermal management strategies. Besides understanding optical characteristics, the lattice dynamical simulations are equally valuable which could help designing the optoelectronic device structures required for improving the emerging needs of high-performance thermoelectric nano- and/or microelectronics.

One must note that only a few theoretical reports exist for studying the structural, electronic, and optical properties of the binary XC materials [8–20]. Limited studies are available that assess the lattice dynamical behavior of C-based alloys [15–20]. However, absolutely no attempts have been made to comprehend the phonon traits of novel strained-layer $(SiC)_m/(GeC)_n$ SLs. By using both experimental and theoretical methods, some improvements are achieved earlier for addressing the necessary constraints in securing progress of the heterostructure-based [21–23] devices. Despite the exploitation of many optical and electrical methods, the estimation of graded interfacial thickness in strained-layer SLs is still a major issue. Previously, high-resolution transmission electron microscopy (HR-TEM) is used to map the distribution of In atoms in $GaN/In_{0.43}Ga_{0.57}N/Al_{0.1}Ga_{0.9}N$ QWs [21]. Irregular distribution of In-based nanoclusters has been detected at the GaN/InN and InGaN/AlGaN interfaces with an estimated diameter of ~1–2 nm. To probe the optical, structural, and electrical attributes of compound semiconductors, it has been a custom-ary tradition to instigate a variety of characterization techniques, including reflectivity, absorption [24–27], high-resolution X-ray diffraction (HR-XRD) [28–34], extended X-ray absorption fine structures (EXAFs) [35–40], photoluminescence (PL) [41–43], secondary ion mass spectroscopy [20–28] (SIMS), spectroscopic ellipsometry (SE) [28], electron-energy loss (EEL) [43,44] measurements, etc. Other experimental methods such as electron param-agnetic resonance (EPR) and far-infrared (FIR) spectroscopy [21–23] are also considered valuable for probing the nature of electronic centers, as well as assessing nanostructured interfacial components by evaluating their vibrational and/or electronic properties [24–44].

In this context, the Raman scattering spectroscopy (RSS) [10,45–54] is considered one of the most trusted techniques for identifying the vibrational characteristics of semiconductor materials. By using RSS, the observed acoustic and optical phonon features in SLs can display strong dependence on the arrangement of their atomic constituents [21,45–47]. While the acoustic part of phonon spectrum is sensitive to the large-scale preparation of SLs [i.e., on the overall periodicity (m, n), capping- and buffer-layer thickness, etc.], the optical phonon spectra depend, however, on their unit cells including individual-layer thickness, and interfacial roughness [45–54]. Previously, several RSS measurements on

GaN/Al$_x$Ga$_{1-x}$N SLs [10,21] have provided strong evidence of the graded alloy interface regions of the order of ~2 nm. Raman studies in (InN)$_m$/(AlN)$_n$, (GaN)$_m$/(AlN)$_n$ and (SiC)$_m$/(Si)$_n$ have also confirmed that: (a) the presence of an interfacial layer thickness may lift optical phonon degeneracy to cause considerable gain in Raman intensity profiles (RIPs), and (b) a suitable choice of layer thickness of the constituent materials can trigger sizable shifts in the optical phonons [10,21]. In novel (SiC)$_m$/(GeC)$_n$ SLs, one could expect observing similar effects due to large separation of the longitudinal and transverse optical ($\omega_{LO}$, $\omega_{TO}$) modes of their constituents despite an overlap of SiC longitudinal acoustic ($\omega_{LA}$) modes on the low-energy optical [12–15] phonon region of GeC. It is, therefore, interesting to undertake methodical lattice dynamical studies [19,54–69] to simulate the phonon properties of (SiC)$_m$/(GeC)$_n$ SLs for comprehending the impacts of m, n and $\Delta$ on the acoustical as well as optical mode frequencies.

The purpose of this paper is (i) to use a modified linear-chain model (M-LCM) [19,54–69] and study the phonon dispersions $\omega_j^{SL}\left(\vec{q}\right)$ of (SiC)$_m$/(GeC)$_n$SLs, (ii) simulate Raman intensities of the graded (SiC)$_{10-\Delta}$/(Si$_{0.5}$Ge$_{0.5}$C)$_\Delta$/(GeC)$_{10-\Delta}$/(Si$_{0.5}$Ge$_{0.5}$C)$_\Delta$ SLs (cf. Sections 2 and 2.1) by meticulously integrating interfacial layer thickness $\Delta$, and (iii) exploit a realistic rigid-ion model (RIM) [54] for calculating the phonon dispersion of zb SiC, zb GeC materials and their SiC)$_m$/(GeC)$_n$ SL$s$ (cf. Sections 2.2 and 2.3) [61–65]. In M-LCM, the SL phonon dispersion calculations are performed by assuming m = n ($\equiv$10) for each binary material. Alloy interfaces are considered in the virtual-crystal approximation by selecting equal proportion of SiC and GeC to describe the interfacial layer thickness, $\Delta$. A bond-polarizability model [19,57], which has provided a very good description of optical phonon features in many superlattices, is adopted here for simulating the RIPs in graded (SiC)$_{10-\Delta}$/(Si$_{0.5}$Ge$_{0.5}$C)$_\Delta$/(GeC)$_{10-\Delta}$/(Si$_{0.5}$Ge$_{0.5}$C)$_\Delta$ SLs (cf. Sections 2 and 2.1–2.3). Appropriate changes in $\Delta$ ($\equiv$1, 2, and 3 MLs) are proven valuable for significantly impacting on the optical phonons of superlattices. The variation in interfacial layer thickness has caused accidental degeneracies and instigated strong localization of modes in the SiC and GeC layers. We strongly believe that these important features are responsible for triggering strong enhancement of RIPs in the middle of the optical phonon frequency region. Besides M-LCM, a realistic rigid-ion model (RIM) [54–68] is also adopted (cf. Sections 3, 3.4 and 3.5) for calculating the phonon dispersions $\omega_j\left(\vec{q}\right)$ of bulk zb SiC and zb GeC materials and $\omega_j^{SL}\left(\vec{q}\right)$ (SiC)$_m$/(GeC)$_n$SLs. A sizable difference in the cation masses of binary materials has caused well-separated optical phonon modes despite a partial overlap of the SiC acoustic ($\omega_{LA}$) modes on the GeC optical phonon [12,15] branches. In the framework of a three-dimensional RIM, we have reported the results of our systematic simulations for SL phonon dispersions $\omega_j^{SL}\left(\vec{q}\right)$ in the growth [001] as well as in-plane directions [100], [110] by carefully considering both the short- and long-range Coulomb interactions. Besides predicting the phonon confinement and anisotropy of optical modes in SLs, the study has also provided strong evidence of acoustic-mode anti-crossing and mini-gap formation. The simulated results are compared and contrasted against the existing experimental/theoretical data with concluding remarks presented in Section 4.

## 2. Theoretical Background

A modified linear chain model is adopted here to describe the vibrational properties, (viz., the phonon dispersions $\omega_j^{SL}\left(\vec{q}\right)$, Raman intensities, etc.) of both the ideal and graded (SiC)$_m$/(GeC)$_n$ SLs. The necessary M-LCM force constants of SLs are obtained by carefully considering interactions up to and including the second nearest neighbors [10,19,54–68]. In the harmonic (cf. Section 3.1) approximation, the equations of motion are obtained and solved by diagonalizing the dynamical matrix to obtain both the eigenvalues and eigenvectors.

### 2.1. Lattice Dynamics

For binary zb SiC and GeC materials, complete phonon dispersions along the high symmetry directions are available [12,14,15]. Appropriate experimental and theoretical phonon frequencies of bulk materials are used for evaluating the necessary M-LCM force constants for calculating $\omega_j^{SL}\left(\vec{q}\right)$ and RIPs [19,54–69]. In $(SiC)_m/(GeC)_n$ SLs, each atom is represented by a plane of atoms, and the associated phonons propagating along the [001] axis are described by creating one-dimensional sets of the equations of motion. The results of phonon dispersions for the sharp interface $(SiC)_m/(GeC)_n$ SLs with m = n = 10 and $\Delta = 0$ are reported in Section 3.1.

### 2.2. Raman Scattering

Raman scattering spectroscopy is a nondestructive method frequently used for describing the lattice dynamics of different types of short-period SLs [10,19–23,54–69] as the measured phonon characteristics exhibit strong dependence on their structural features. The RSS has been applied to many II-VI, III-V, and IV-IV MQWs and SLs [10,19–23,54–69]. In the bulk zb structured materials, the Raman scattering selection rule forbids $\omega_{TO}$ modes in the backscattering geometry of (001) face [58]. On the other hand, the symmetry arguments have suggested that only $\omega_{LO}$ modes can play important roles in SLs. By using the linear-chain models in the framework of bond polarizability methods, many successful studies exist [19–23,54–69] on SLs for comprehending their experimental Raman spectra in the optical phonon frequency region.

Raman Intensity Profiles

Following Zhu and Chao [57] and adopting a bond polarizability method in the framework of M-LCM, we have simulated phonon dispersions $\omega_j^{SL}\left(\vec{q}\right)$ (cf. Section 3.1), Raman intensities $I(\omega)$ (cf. Section 3.2), and atomic displacements $u_j^{SL}\left(\vec{q}\right)$ (cf. Section 3.3) for both the ideal and graded strained layer $(SiC)_{10-\Delta}/(Si_{0.5}Ge_{0.5}C)_\Delta/(GeC)_{10-\Delta}/(Si_{0.5}Ge_{0.5}C)_\Delta$ SLs. The calculations of Raman intensities are performed by using [57]

$$I_{xx}\,(\omega)\ \propto |\sum\nolimits_A \alpha_{xx,A}(u_{1z} - u_{3z})|^2, \tag{1a}$$

for modes of $A_1$-type symmetry and

$$I_{xy}\,(\omega)\ \propto |\sum\nolimits_A \alpha_{xy,A}(u_{1z} + u_{3z} - 2u_{0z})|^2, \tag{1b}$$

for modes having $B_2$ symmetry. To calculate $I(\omega)$ for SiC/GeC SLs using Equation (1), we have assumed fixed values of the polarizability constants $\alpha_{ij,A}$ throughout. The summation in Equation (1) over all A atoms are represented either by Si or Ge, with $u_0$ referring to the displacements for each of these, while $u_1$ and $u_3$ represent the displacements of their nearest-neighbor C atoms.

### 2.3. Rigid-Ion-Model for Bulk Binary Materials

A realistic three-dimensional RIM [54] that we adopted here to study the lattice dynamics of short-period $(SiC)_m/(GeC)_n$ SLs has been applied before for comprehending the phonon characteristics of both perfect and imperfect semiconductors [64]. For binary zb SiC and zb GeC materials of tetrahedral $T_d$ point group symmetry, the phonon dispersions $\omega_j\left(\vec{q}\right)$ are obtained by solving the equations of motion for the ions [54]:

$$\omega_j^2\left(\vec{q}\right)e_\alpha(\kappa|\vec{q}j) - \sum\nolimits_{\kappa'\beta} D_{\alpha\beta}^{sC}\left(\kappa\kappa'\Big|\vec{q}\right)e_\beta(\kappa'|\vec{q}j); \ \kappa, \kappa' = 1, 2, \tag{2}$$

where $D_{\alpha\beta}^{sC}\left(\kappa\kappa\prime\Big|\vec{q}\right)\left[\equiv D_{\alpha\beta}^{s}\left(\kappa\kappa\prime\Big|\vec{q}\right)+D_{\alpha\beta}^{C}\left(\kappa\kappa\prime\Big|\vec{q}\right)\right]$ signifies the elements of dynamical matrices involving both the short-range ($D_{\alpha\beta}^{s}\left(\kappa\kappa\prime\Big|\vec{q}\right)$) and long-range Coulomb ($D_{\alpha\beta}^{C}\left(\kappa\kappa\prime\Big|\vec{q}\right)$) interactions.

The $D_{\alpha\beta}^{s}\left(\kappa\kappa\prime\Big|\vec{q}\right)$) matrix includes two (A, B) nearest-neighbor cation–anion force constants, eight ($C_i$, $D_i$, $E_i$, and $F_i$ with i = 1, 2) next-nearest-neighbor cation–cation and anion–anion interactions. A parameter related to the electron transfer charge ($Z_{eff}$) between ions describes the strength of long-range Coulomb interaction, $D_{\alpha\beta}^{C}\left(\kappa\kappa\prime\Big|\vec{q}\right)$. To simulate the phonon dispersions $\omega_j\left(\vec{q}\right)$ of binary zb SiC and GeC materials, we have carefully optimized the short- and long-range interactions following the successive nonlinear least square fitting procedures [65]. The eigenvalue equation (Equation (2)) is solved numerically for each $\vec{q}$-vector points in the Brillouin zone (BZ) to obtain (cf. Section 3) the phonon dispersions $\omega_j\left(\vec{q}\right)$ and one-phonon density of states (DOS) $g(\omega)$ of the binary compounds.

### 2.3.1. Rigid-Ion Model for $(SiC)_m/(GeC)_n$ Superlattices

Our method for studying the phonon dispersions of $(SiC)_m/(GeC)_n$ SLs uses a RIM developed by Kunc [54] for the bulk zb materials. The model has been successfully applied to comprehend the role of isolated and complex defect centers in several imperfect semiconductors [64] and extended to SLs by others [66–68]. In particular, the works of Molinari et al. [66], Ren et al. [67], and Kanellis [68] have provided the detailed accounts of simulating the phonon characteristics of III-V GaAs/AlAs SLs.

For the pseudo-morphic $(SiC)_m/(GeC)_n$ (001) SL of tetragonal $D_{2d}$ point-group symmetry, we have considered zb SiC and GeC binary materials of m and n bilayers, respectively, and stacked them alternatively along the [001] growth direction. The atomic basis of SL is defined as a bulk crystal having 2 (m + n) atoms. As the symmetry of SL is lowered than the bulk materials, there are more internal degrees of freedom in the atomic arrangements of the superlattices. The large atomic basis results in the reduction of the SL Brillouin zone (BZ) and triggers folding of phonons along the growth direction. To simulate the phonon features of $(SiC)_m/(GeC)_n$ SLs by using a RIM, we treated each plane normal to the growth direction with one type of ions as a "layer", identifying it by using a label s; a combination of two adjacent layers as a "bilayer" with $\chi$; and a sublattice as a collection of equivalent atomic layers, one in each period of the SL. The two layers are designated by $\sigma$ ($\equiv$1, 2) such that $\chi$ is equivalent to a composite index (s, $\sigma$). In this scheme, we assumed the short-range interatomic force constants to be the same as those of the binary zb SiC and zb GeC materials except for the interactions between Si and Ge atoms across the interface, that is, when the two atoms are positioned in different layers. In this situation, we have evaluated the short-range force constants by taking an average of the Si–Si interaction in the bulk zb SiC and the Ge–Ge interaction in the bulk zb GeC.

### 2.3.2. Coulomb Interactions in Superlattices

Unlike binary zb materials, the calculations of long-range Coulomb interactions in SLs become rather cumbersome due to the lowered symmetry. However, it is extremely imperative to include the long-range Coulomb interactions for defining the dynamical matrix of $(SiC)_m/(GeC)_n$ SLs. The inclusion of Coulomb interactions is particularly important for accomplishing not only the correct anti-crossing, and confinement behavior in the acoustic phonon frequency range but also for attaining the confinement and anisotropic results of phonons in the optical phonon frequency region. We have used the Ewald method following Ren et al. [67] and calculated the Coulomb matrix elements $D_{\alpha\beta}^{C}\left(\chi\chi\prime\Big|\vec{q}\right)$ in SLs between s and s$\prime$ sublattices as $\vec{q} \to 0$ by using [67]

$$D_{\alpha\beta}^C\left(\chi\chi'\Big|\vec{q}\right)) = \frac{4\pi Z_s Z_{s'}}{\Omega N_p}\left(\frac{q_\alpha q_\beta}{\vec{q}^2} - \delta_{\alpha z}\delta_{\beta z}\right) + D_{\alpha\beta}^C(s,s'), \qquad (3)$$

where $\alpha$ and $\beta$ indicate the x, y, and z directions; $Z_s$ and $Z_{s'}$ signify the atomic-transfer charges of atoms in the s and s′ sublattices, respectively; $\Omega$ is the volume of a unit cell of bulk material; and $N_p$ represents the total number of bilayers in an SL period. The function $\frac{q_\alpha q_\beta}{\vec{q}^2}$ on the right-hand side of the first part of Equation (3) attains different values when $\vec{q}$ approaches zero from diverse directions. The first term arises from macroscopic electric field due to the presence of dipoles at the atomic sites, while the second term $D_{\alpha\beta}^C(s,s\prime)$ is independent on the direction. The results of $(SiC)_m/(GeC)_n$ short-period SLs are reported in Sections 3.4 and 3.5 both in the growth direction [001] as well as in directions [100] and [110] perpendicular to it.

## 3. Numerical Computation Results and Discussions

For $(SiC)_m/(GeC)_n$ SLs, we have reported the results of our methodical calculations of the phonon dispersions $\omega_j^{SL}\left(\vec{q}\right)$, RIPs and atomic displacements $u_j^{SL}\left(\vec{q}\right)$ by exploiting a phenomenological M-LCM (Sections 3.1–3.3) approach. A realistic RIM [54] (cf. Sections 3.4 and 3.5) is used for simulating the dynamical properties of SLs along the growth direction [001] as well as in directions [100] and [110] perpendicular to it. Theoretical results are compared/contrasted and discussed with the existing experimental and theoretical data in Section 4.

### 3.1. M-LCM Phonon Dispersions of $(SiC)_m/(GeC)_n$

In Figure 1, we present our M-LCM results of the phonon dispersions $\omega_j^{SL}\left(\vec{q}\right)$ for a $(SiC)_m/(GeC)_n$ (001) SL with an ideal (or sharp) interface. With m = n = 10, the SL modes consist of 20 optical and 20 acoustical phonon branches. In this context, the 40 vibrational modes are identified by using numbers with increasing frequencies. From Figure 1, it is obvious that the folded acoustic phonons (indicated by blue-colored dash lines) are highly dispersive for 1–14 modes, while the remaining 6 branches are nondispersive, as usually seen for the optical phonons (displayed with full red (GeC-like) and green (SiC-like) lines). The impact of interfacial layer thickness $\Delta$ ($\equiv$1, 2, and 3 MLs) on the phonon dispersions of graded $(SiC)_{10-\Delta}/(Si_{0.5}Ge_{0.5}C)_\Delta/(GeC)_{10-\Delta}/(Si_{0.5}Ge_{0.5}C)_\Delta$ SLs is also carefully studied. By increasing $\Delta$ from 1 to 3 MLs, the acoustical phonon dispersions revealed insignificant changes (<~3 cm$^{-1}$). However, the frequencies of optical mode branches are considerably affected. For instance, the high-frequency SiC-like #33–34 (low-frequency GeC-like #28) optical modes substantially shifted downward (upward) up to ~47 cm$^{-1}$ with $\Delta$ = 3. From the calculated RIPs and atomic displacements (cf. Sections 3.4 and 3.5), the impact of interfacing $\Delta$ is clearly conspicuous, causing strong localization of the confined optical modes (COMs).

### 3.2. Raman Scattering Profiles in SLs

The effective use of periodic $(GaN)_m/(AlN)_n$ (001) SLs (for instance) has required a detailed theoretical study of their fundamental physical properties, as well as the development of new quantitative experimental diagnostics to improve their growth technology [10,21,57–60,69]. In this respect, Raman scattering spectroscopy is recognized as a versatile tool for the nondestructive study of the phonon spectrum in SLs by using a high spatial resolution [21,57,69]. Many theoretical studies of calculating the RIPs of III-V-based and IV-IV-based SLs have used the linear-chain models by integrating the bond-polarizability schemes for interpreting the experimental Raman scattering data [18,19,57,69].

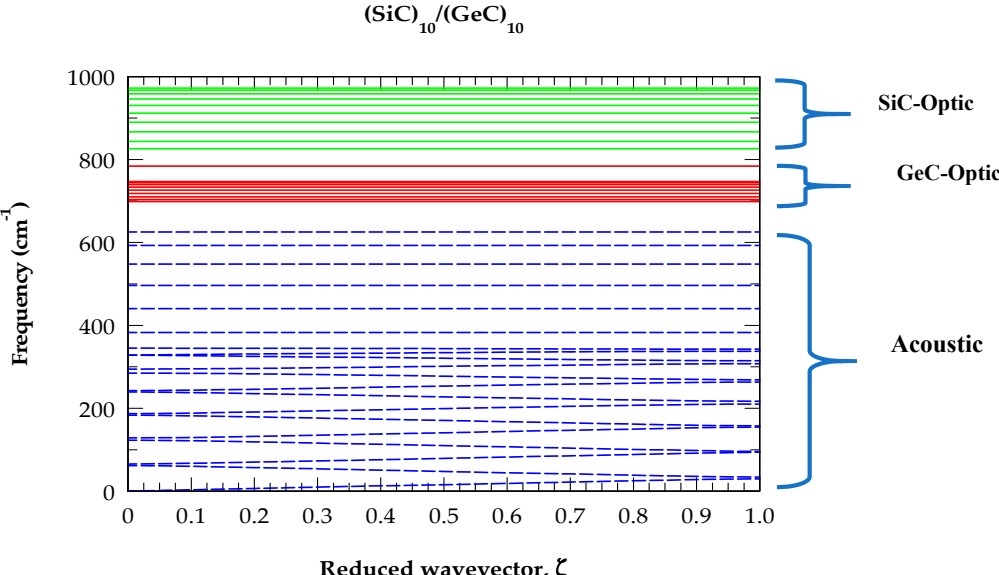

**Figure 1.** Calculated phonon dispersion curves of an ideal $(SiC)_{10}/(GeC)_{10}$ SL based on a modified linear-chain model. The dotted blue color lines represent the acoustic phonons. The green and red color full lines, respectively, represent zb SiC- and zb GeC-like confined optical modes (see text).

Here, we have adopted a methodology proposed by Zhu and Chao [57] and reported the results of our calculations of Raman scattering profiles for both the ideal, $\Delta = 0$, and graded strained layer $(SiC)_{10-\Delta}/(Si_{0.5}Ge_{0.5}C)_{\Delta}/(GeC)_{10-\Delta}/(Si_{0.5}Ge_{0.5}C)_{\Delta}$ SLs by assuming the polarizability constant values as unchanged throughout. The validation of our study is to emphasize only the optical phonon frequency region of the SLs and closely monitor the changes caused by varying m, n, and $\Delta$. In Figure 2a,b, the impact of m and n on the simulated Raman intensities is displayed in the range of $650 \text{ cm}^{-1} \leq \omega \leq 1050 \text{ cm}^{-1}$ for the ideal $(SiC)_m/(GeC)_n$ (001) SL (or with sharp interface $\Delta = 0$, considering m (n) = 10 and varying n (m) from 2 to 10. The effects of $\Delta$ ($\equiv 0$, 1, 2, and 3 MLs) on Raman intensities in graded SLs with m (=n = 10) are also reported (see Figure 3a).

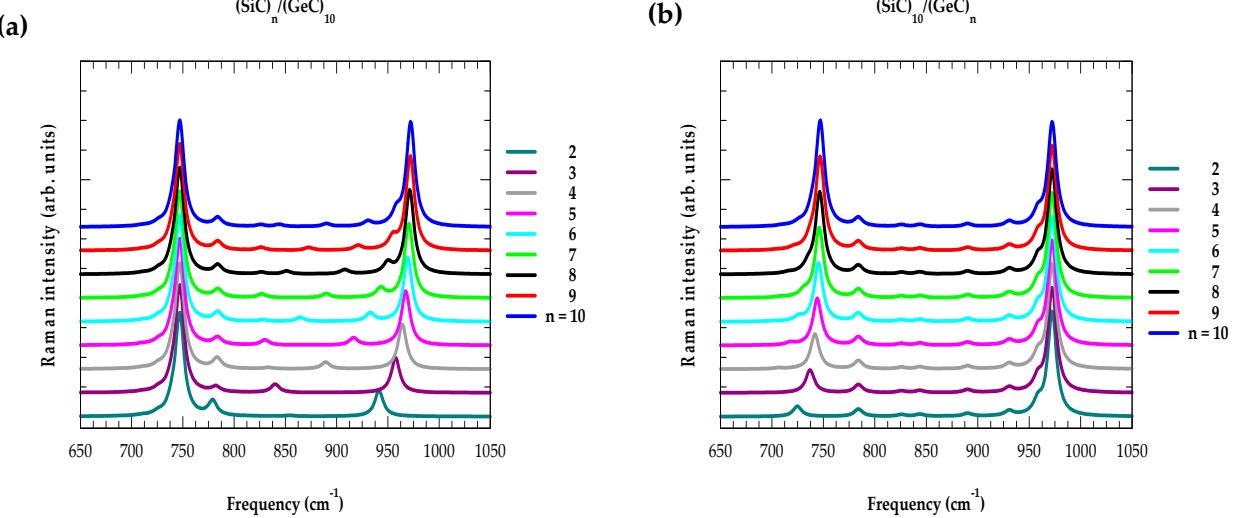

**Figure 2.** (**a**) Calculated variation in Raman intensity profiles for $(SiC)_n/(GeC)_{10}$ superlattices by changing SiC-layer thickness n from 2 to 10 (see text); (**b**) calculated variation in Raman intensity profiles for $(SiC)_{10}/(GeC)_n$ superlattices by changing GeC layer thickness n from 2 to 10 (see text).

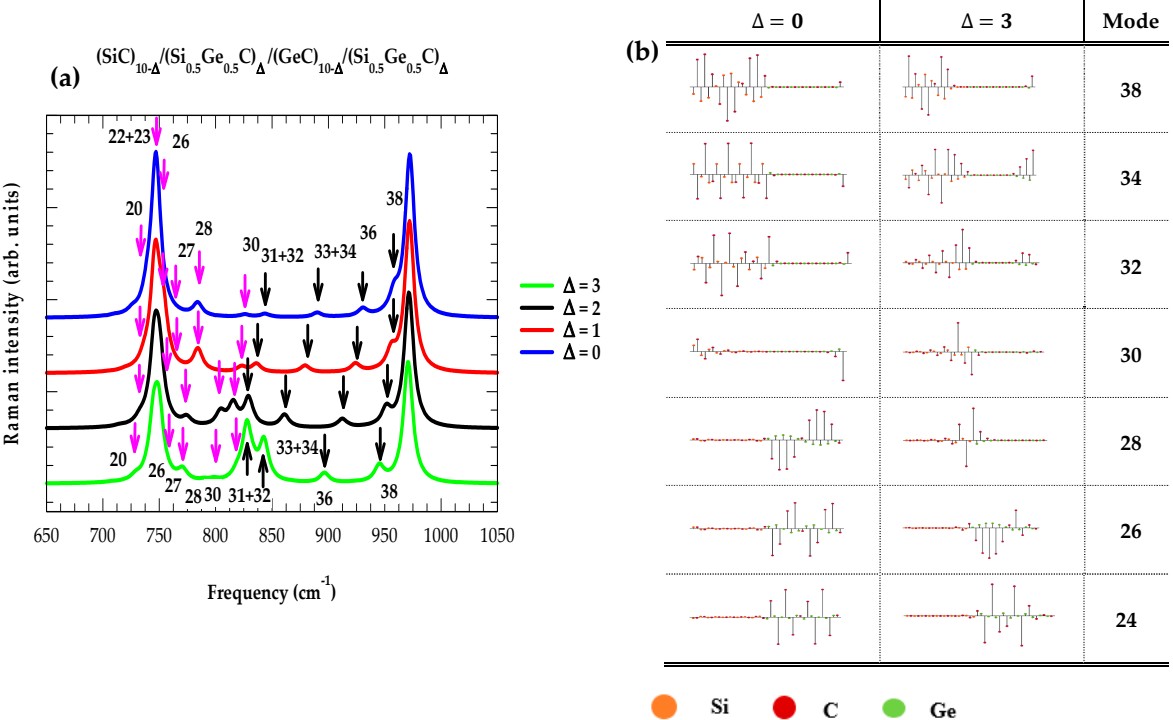

**Figure 3.** (**a**) Calculated variation in the mode frequencies and their corresponding Raman intensity in graded $(SiC)_{10-\Delta}/(Si_{0.5}Ge_{0.5}C)_{\Delta}/(GeC)_{10-\Delta}/(Si_{0.5}Ge_{0.5}C)_{\Delta}$ SL as a function of interfacial layer thickness $\Delta$ ($\equiv$0, 1, 2, and 3 MLs); (**b**) calculated atomic displacements of selected SiC- and GeC-like optical modes for the graded $(SiC)_{10-\Delta}/(Si_{0.5}Ge_{0.5}C)_{\Delta}/(GeC)_{10-\Delta}/(Si_{0.5}Ge_{0.5}C)_{\Delta}$ SL with $\Delta = 0$ and 3. Atomic displacements of Si, C, and Ge atoms are indicated using orange, violet, and green circles (see text).

For m (=n = 10), the perusal of Figure 2a,b has revealed two dominant features: (i) one related to the zb GeC-like $\omega_{LO}$ phonon at 747.2 cm$^{-1}$ (#22–23) and (ii) the other linked to the zb SiC-like $\omega_{LO}$ mode at 977.6 cm$^{-1}$ (#40). Even though the sharp Raman feature labeled as bands #22–23 has a frequency coincident with that of the bulk zb GeC-like $\omega_{LO}$ phonon, it does not correspond to any of the zb GeC-like COMs in SL. Besides the two major bands, there are several weak Raman intensity features located at frequencies $\omega \equiv 752$ cm$^{-1}$, 763 cm$^{-1}$, 782 cm$^{-1}$, 825 cm$^{-1}$, 844 cm$^{-1}$, 890 cm$^{-1}$, 931 cm$^{-1}$, and 958 cm$^{-1}$, which are associated with the phonon modes #26, 27, 28, 30, 31 + 32 (quasi-degenerate), 33 + 34 (quasi-degenerate), 36, and 38, respectively. We have identified these Raman structures by comparing their intensities with the calculated M-LCM phonon frequencies of ideal $(SiC)_{10}/(GeC)_{10}$ SL (cf. Figure 1). In Figure 2a,b, we report our results of simulated Raman intensities by carefully varying n from 2 to 10 in $(SiC)_n/(GeC)_{10}$ and $(SiC)_{10}/(GeC)_n$ SLs, respectively. In these studies, although the zb GeC- and zb SiC-like features remained nearly constant, we have noticed significant shifts in the zb SiC- and zb GeC-like mode frequencies as well as their intensities. For instance, by varying n in $(SiC)_n/(GeC)_{10}$ SLs, the calculated zb SiC-like bands exhibited a steady blue shift with the decrease in Raman intensities. In contrast, by changing n in the $(SiC)_{10}/(GeC)_n$ SLs, we observed somewhat inconsistent modifications in the zb GeC-like Raman intensities as well as the shifts in their mode frequencies. One must note that similar shifts in the optical phonon frequencies and intensities have been experimentally observed in recent Raman scattering studies of $(GaN)_m/(AlN)_n$ SLs [10,69]. In Figure 3a, the simulated results of RIPs are reported for the graded $(SiC)_{10}/(GeC)_{10}$ SLs by carefully varying the interfacial layer thickness $\Delta$ ($\equiv$0–3 MLs). In the ideal situation (i.e., $\Delta$ ($\equiv$0)), we noticed that the optical modes between #22 and 29 remained confined to the zb GeC layers, and mode #30 is an interfacial GeC-SiC-like mode, while the remaining modes >31 and $\leq$40 are restricted to the zb SiC

layers. As $\Delta$ ($\equiv$1, 2, and 3 MLs) changed, significant modifications are revealed in the simulated phonons and RIPs, especially in the middle of the optical mode frequency region. For $\Delta$ ($\equiv$3 MLs), the zb GeC-like mode #24 and SiC-like mode #38 exhibited only a slight variation in frequency and intensity.

However, the phonon energy and intensity of zb GeC-like mode #28 (zb SiC-like #31–32, 33–34 modes) moved upward (downward) with the increase of $\Delta$. With $\Delta$ ($\equiv$3 MLs), the GeC-(SiC)-related #28 (#31 + 32, #33–34 quasi-degenerate) modes shifted upward (downward) by ~47 cm$^{-1}$, became localized, and prompted significant enhancement in the Raman intensities. By including the interfacial thickness using a random element iso-displacement model in the (GaN)$_m$/(AlN)$_n$ (001) SLs, similar observations have been recently reported by Davydov et al. [10,69].

### 3.3. Atomic Displacements in Superlattices

As shown in Figure 3a, the impact of increasing $\Delta$ ($\equiv$1, 2, and 3) is reflected in a gradual improvement in Raman intensities in the middle of the optical phonon frequency region. This increase in intensities (Figure 3a) can be understood by examining the atomic displacements of the apposite phonon modes. For $\Delta$ ($\equiv$0, 3 MLs), the results of our calculations for atomic displacements are shown in Figure 3b for a selected number of modes between #24 and #38. The atomic displacements of Si, C, and Ge atoms in the (SiC)$_{10}$/(GeC)$_{10}$ SL are indicated by using orange, violet, and green colored circles, respectively. With respect to an ideal situation $\Delta$ ($\equiv$0), the changes in atomic displacements for various phonon modes by increasing $\Delta$ ($\equiv$3) have strongly corroborated the enhancement observed in Raman intensity features (see Figure 3a) for the graded (SiC)$_{10}$/(GeC)$_{10}$ SLs.

### 3.4. RIM Calculations of Bulk zb SiC and zb GeC

By exploiting a realistic RIM and using the optimized set of force constants, we havecalculated the phonon dispersions $\omega_j\left(\vec{q}\right)$ (see Figure 4a) and one-phonon density of states (DOS) (see Figure 4b) for the bulk zb SiC and zb GeC materials. The simulated results of $\omega_j\left(\vec{q}\right)$ reported in Figure 4a along the high-symmetry directions ($\Gamma \rightarrow X \rightarrow K \rightarrow \Gamma \rightarrow L \rightarrow X \rightarrow W \rightarrow L$) are compared/contrasted reasonably well with the existing zb SiC inelastic X-ray scattering data [14] (indicated by symbols) and the theoretical data [15].

**(a)**                                                **(b)**

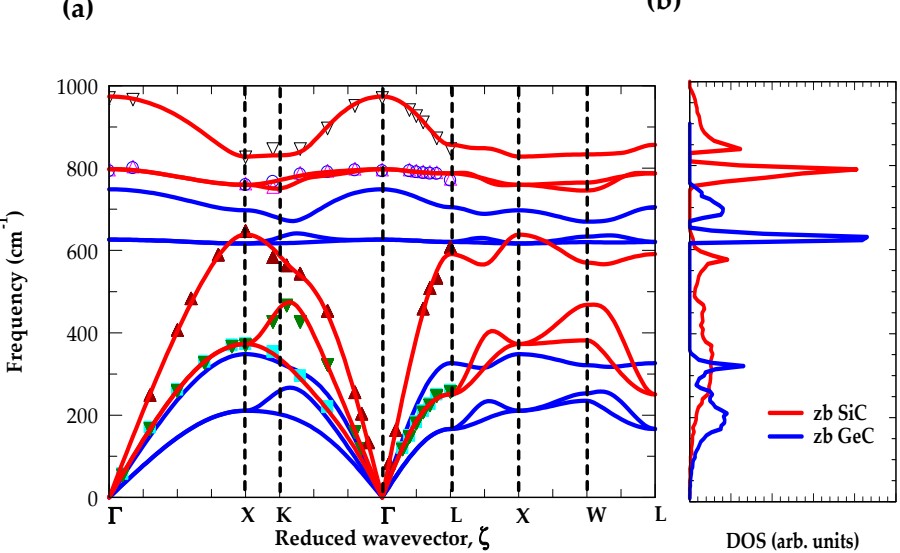

**Figure 4.** (**a**) Comparison of the RIM calculated phonon dispersions for zb SiC (red lines) and zb GeC (blue lines) with the existing experimental (inelastic X-ray scattering data of zb-SiC (symbols) [14] and zb GeC theoretical data [15]; (**b**) calculated RIM results of one-phonon density of states for zb SiC (red lines) and zb GeC (blue lines) (see text).

From Figure 4a, we noticed that the acoustic and optical phonon branches are well separated. Clearly, these results have suggested that the heavier Si and Ge atoms contribute to the acoustical modes, while the lighter C atoms affect only the optical phonons. It should be noted that proceeding from the Ge to Si atoms (i.e., from the higher to lower atomic number), the acoustic and optical phonon frequencies of zb SiC steadily increased relative to the zb GeC binary compound. This fact is also reflected in our calculations of the one-phonon DOS (see Figure 4b), where a large phonon gap is observed between the acoustic and optical branches of zb GeC (355–620 cm$^{-1}$) [15] as compared to a small phonon gap in the zb SiC (615–750 cm$^{-1}$) [14]. These results of $\omega_j\left(\vec{q}\right)$ and one-phonon DOS lead us to two compelling outcomes: (a) The large splitting of $\omega_{LO}$ and $\omega_{TO}$ in zb SiC and zb GeC strongly suggests that the long-range Coulomb interactions in SLs cannot be ignored, and (b) as the $\omega_{LA}$ modes of zb SiC partially overlap on the optical zb GeC bands, the position of folded acoustic phonons in $(SiC)_m/(GeC)_n$ could be sensitive to the period of the SL (cf. Sections 3.5.1–3.5.3). Again, the correct values of Debye temperature $\Theta_D(T)$ and specific heat $C_v(T)$ reported at different T for the zb SiC and GeC (Section 3.4.1) materials have provided additional support to our choice of the RIM for calculating the phonon dispersions of $(SiC)_m/(GeC)_n$ SLs.

### 3.4.1. Debye Temperature and Specific Heat of Bulk zb SiC and zb GeC

The rigid-ion model calculations of $\Theta_D(T)$ and $C_v(T)$ for the binary zb SiC and zb GeC bulk materials are reported in Figure 5a,b, respectively. Comparison with the existing experimental [25] and theoretical [15] data is reasonably good. At higher temperatures, the small discrepancies between the experimental and simulated values of $\Theta_D(T)$ in zb SiC are not considered to be as meaningful as the deviations suggest (see Table 1).

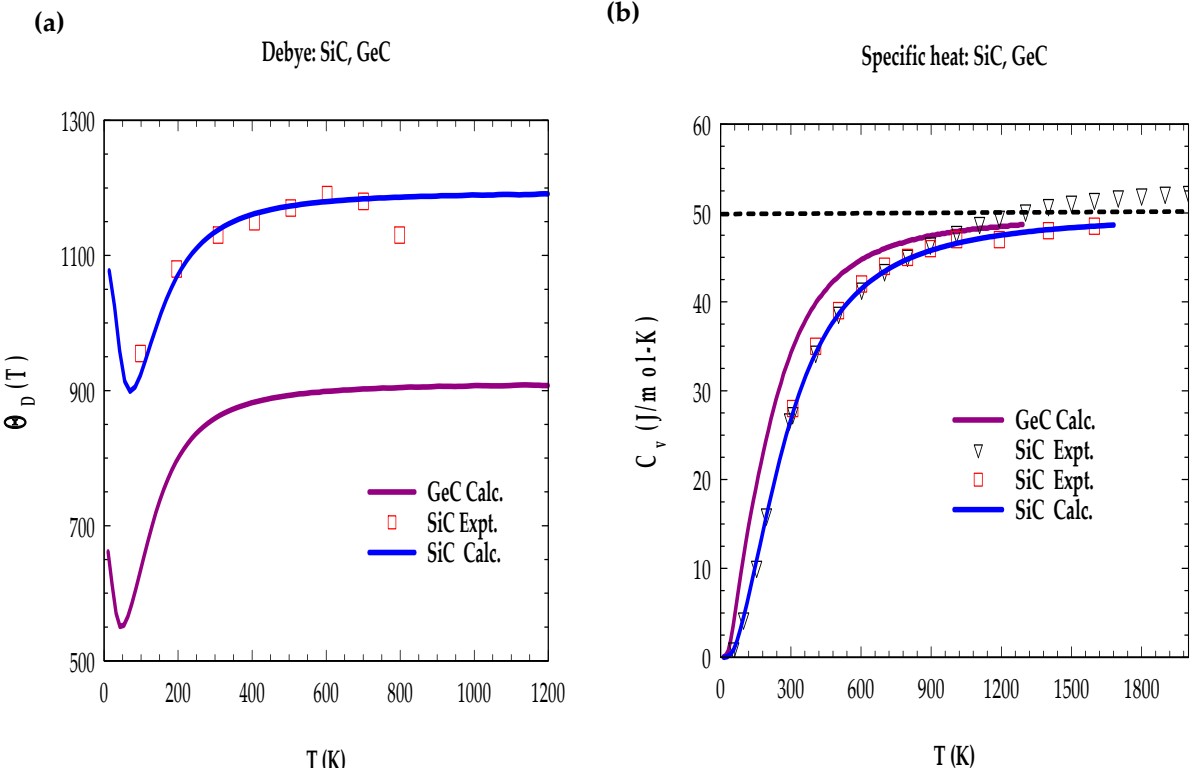

**Figure 5.** (**a**) Calculated Debye temperature $\Theta_D$ as a function of T based on rigid-ion model for SiC (blue line) and GeC (red line) compared with experimental data [25] (symbols) and theoretical data [15,26]; (**b**) same as (**a**) but for the specific heat $C_v(T)$ as a function of T for SiC and GeC (see text).

**Table 1.** The relevant set of physical parameters for SiC and GaC used in optimizing the rigid-ion model parameters. The elastic constants $c_{11}$, $c_{12}$, and $c_{44}$ are in $10^{11}$ dyn cm$^{-2}$; lattice constant $a_0$ in (Å) and specific heat $C_v$ in (J/mol-K). The critical-point phonon frequencies are in wave number (cm$^{-1}$).

| Physical Parameter | Expt. | SiC Theory | Our | GeC Theory | Our |
|---|---|---|---|---|---|
| $c_{11}$ | 39.0 [a] | 38.5–39.0 [e] | 38.7 [f] | 31.8 [e] | 31.9 [f] |
| $c_{12}$ | 14.2 [a] | 13.2–14.2 [e] | 14.3 [f] | 10.4 [e] | 10.7 [f] |
| $c_{44}$ | 25.6 [a] | 24.3–25.6 [e] | 25.8 [f] | 19.2 [e] | 17.6 [f] |
| $a_0$ | 4.36 [a] | 4.28–4.40 [e] | 4.36 [f] | 4.59 [e] | 4.59 [f] |
| LO($\Gamma$) | 972 [b] | 945–956 [e] | 974 [f] | 748 [e] | 749 [f] |
| TO($\Gamma$) | 796 [b] | 774–783 [e] | 797 [f] | 626 [e] | 626 [f] |
| LO(X) | 829 [b] | 807–829 [e] | 828 [f] | 697 [e] | 698 [f] |
| TO(X) | 761 [b] | 741–755 [e] | 760 [f] | 617 [e] | 617 [f] |
| LA(X) | 640 [b] | 622–629 [e] | 639 [f] | 348 [e] | 348 [f] |
| TA(X) | 373 [b] | 361–366 [e] | 373 [f] | 214 [e] | 210 [f] |
| LO(L) | 838 [b] | 817–838 [e] | 857 [f] | 705 [e] | 705 [f] |
| TO(L) | 766 [b] | 747–766 [e] | 787 [f] | 612 [e] | 621 [f] |
| LA(L) | 610 [b] | 601–610 [e] | 591 [f] | 331 [e] | 326 [f] |
| TA(L) | 266 [b] | 257–261 [e] | 251 [f] | 162 [e] | 166 [f] |
| $\theta_D(T \rightarrow 0)$ | 1080 [c] | | 1080 [f] | | 670 [f] |
| $\theta_D(T \rightarrow 800)$ | 1270 [d] | | 1186 [f] | | 904 [f] |
| $C_v(T \rightarrow 298)$ | 26.84 [d] | | 26.24 [f] | | 34.3 [f] |

(a) Ref. [31]; (b) Ref. [14]; (c) Ref. [25]; (d) Ref. [26]; (e) Ref. [15]; (f) our study.

Since the measured data in [25] do not present the uncertainties, due to the difficulties of estimating small changes in $C_v$ at higher temperatures, we believe that the expected ambiguities in the appraised values of $\Theta_D(T)$ could be sizable. Overall, the favorable comparison of phonon dispersions, DOS, Debye temperatures $\Theta_D(T)$, and specific heats $C_v$ (T) for zb SiC and zb GeC [15] (Table 1) confirms the accuracy of our choice of RIM for extending it to comprehend the lattice dynamical characteristics (Section 3.5) of $(SiC)_m/(GeC)_n$ SLs.

*3.5. RIM Calculations of $(SiC)_m/(GeC)_n$ SLs*

Before presenting our results of RIM calculations on the phonon dispersions for the short-period $(SiC)_m/(GeC)_n$ [001] SLs, one must note that, in the $D_{2d}$ point group, the symmetry of zone-center longitudinal modes is $B_2$ and E for the doubly degenerate transverse phonons. If the wave vector is parallel ($q_{||}$) to the SL axis (direction $\Gamma$–Z), the point group of wave vector is $C_{2v}$, and the mode symmetries are $A_1$ for the longitudinal and $B_1$ and $B_2$ for the transverse phonons. For wave vectors perpendicular ($q_\perp$) to the SL axis (x-axis: direction $\Gamma$-X), and for small $\vec{q}'$s ($\left|\vec{q}'\right| \rightarrow 0$) forming an angle θ with the SL axis, the point group of wave vector becomes $C_s$, and the zone-center $B_2$ modes transform into $A'$ and $A''$, respectively. The $A''$ modes are transverse in nature, with ions vibrating along the y direction having a constant frequency, i.e., independent of the angle θ. The $A'$ modes, on the other hand, have mixed polarization, showing dispersions with the direction of the wave vector (or angle θ), especially for the optical phonons, due to the macroscopic field associated with them. Only the anti-symmetric (odd-numbered) modes accompany the macroscopic field, while it vanishes for the symmetric (even-numbered) phonons.

### 3.5.1. Phonons of SLs along the Growth Direction

Figure 6a illustrates the results of our RIM phonon dispersions for a $(SiC)_2/(GeC)_2$ SL along the growth direction [001]. For comparison, the calculations reported in Figure 6b,c are for the bulk phonon dispersions $\omega_j\left(\vec{q}\right)$ of the zb SiC and zb GeC along the [001] $\Gamma$-X direction, with folded modes over the same period indicated by vertical black dotted lines. Although the optical phonon frequencies of the bulk zb SiC and zb GeC are well separated, the SiC $\omega_{LA}$ modes overlap with the optical branch of GeC. This is clearly visible in the $\omega_j\left(\vec{q}\right)$ dispersions (see Figure 4a). Consequently, one expects the SL phonon dispersions $\omega_j^{SL}\left(\vec{q}\right)$ derived from the bulk SiC optical phonons to remain confined in the zb SiC layers, i.e., COMs. On the other hand, the mixing of optical zb GeC phonons with zb SiC $\omega_{LA}$ phonons may create quasi-confined optical modes (Q-COMs) in the zb GeC layers. Again, it is possible to determine the degree of confinement from the smallest imaginary solution to the corresponding bulk lattice dynamical equations at appropriate frequencies.

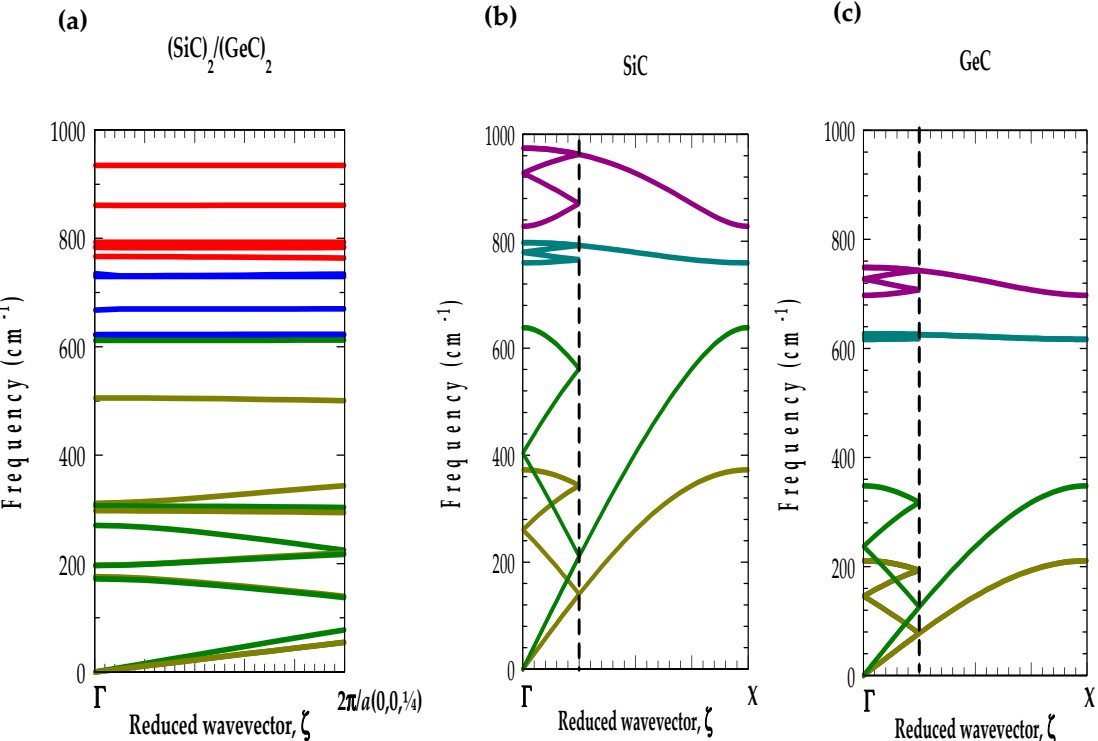

**Figure 6.** (**a**) The phonon dispersion curves of a $(SiC)_2/(GeC)_2$ superlattice along the [001] direction; (**b**) phonon dispersion curves of SiC along the [001] $\Gamma$-X direction folded over the same period as that of the superlattice (indicated by vertical black dashed lines); (**c**) phonon dispersion curves of GeC along the [001] $\Gamma$-X direction folded over the same period as that of the superlattice (indicated by vertical black dashed lines).

### 3.5.2. Phonons of SLs Perpendicular to the Growth Direction

In Figure 7a, we have reported the results of our RIM calculations of phonon dispersions for a $(SiC)_2/(GeC)_2$ SL in the directions [110] (left panel) and [100] (right panel) perpendicular to the growth axis [001]. Like M-LCM (Figure 1), the phonon frequencies $<600$ cm$^{-1}$ (Figure 6a) are derived from the acoustical branches of zb SiC and zb GeC, which exhibit dispersive behavior. The higher optical phonons of zb GeC (blue lines) and zb SiC (red lines), however, reveal dispersion-less characteristics. Comparing the results of Figure 6a with Figure 7a, one may note that the calculated optical modes are distinctly different near the $\left|\vec{q}\right| \to 0$ from the growth and in-plane directions. This means that the phonons are anisotropic functions of $\vec{q}$ at $\left|\vec{q}\right| = 0$. On the other hand, it can be seen from

Figure 7a that the phonon frequencies remain unchanged for $\vec{q}$ approaching zero from different in-plane directions. In Figure 7b, the complete phonon dispersions $\omega_j^{SL}\left(\vec{q}\right)$ are displayed for the $(SiC)_2/(GeC)_2$ superlattice along the high symmetry $\Gamma \rightarrow X \rightarrow M \rightarrow X'$ $\rightarrow \Gamma$ directions. Due to zone-folding effects, the acoustical modes of SLs exhibited strong mixing of the bulk $\omega_{TA}$, $\omega_{LA}$ phonons and caused complications in the dispersion curves with the appearance of several stop bands at some finite values of $\vec{q}$. The stop bands of acoustical phonons for oblique incidence have been experimentally observed [70,71] and studied using an elastic theory [71].

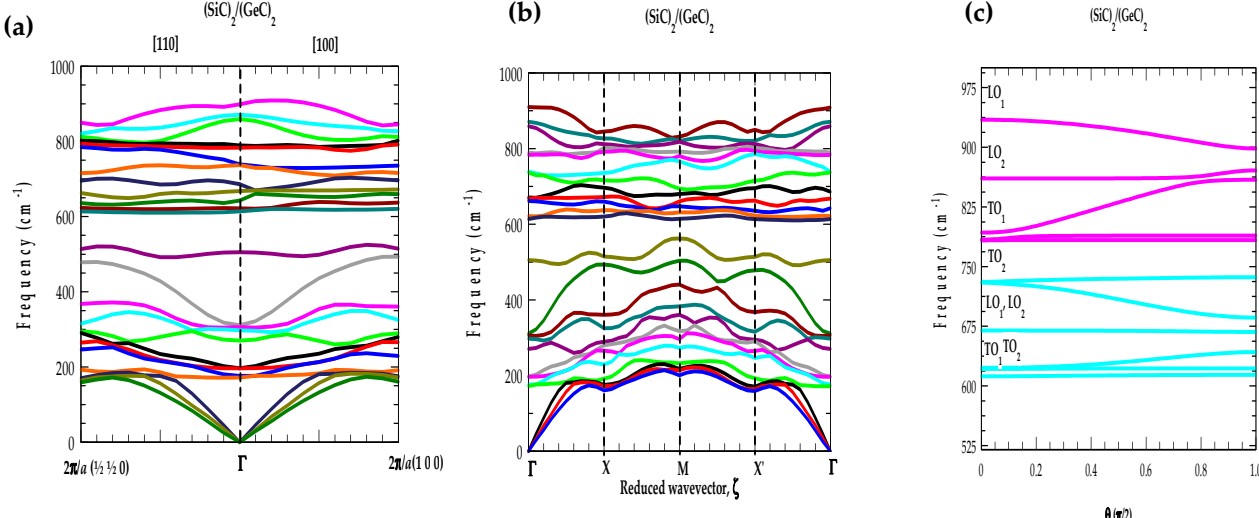

**Figure 7.** (**a**) Phonon dispersion curves of $(SiC)_2/(GeC)_2$ superlattice along the [110] (left panel) and [100] direction (right panel); (**b**) complete phonon dispersion curves of $(SiC)_2/(GeC)_2$ superlattice along the $\Gamma - X - M - X' - \Gamma$ direction in the superlattice Brillouin zone; (**c**) frequencies of the zone-center ($\vec{q} = 0$) optical modes for a $(SiC)_2/(GeC)_2$ superlattice as a function of θ (see text).

### 3.5.3. Angular Dependence of Optical Phonons in SLs

In Figure 7c, we have reported our results for the angular dependence of zone-center optical phonons in $(SiC)_2/(GeC)_2$ SL, where the angle θ of the wavevector $\vec{q}$ is measured from the growth direction. The value of θ changes from 0 to $\pi/2$ as $\vec{q}$ varies from the growth [001] to [100] direction in a plane normal to the [010]. Clearly, the perusal of Figure 7c reveals two sets of vibrational modes, one is associated with the SiC-like phonons (the upper-frequency set), indicated with magenta lines, and the other GeC-like modes (the lower-frequency set), shown with sky blue lines. As stated before, based on the angular momentum considerations, only the odd number of optical phonons exhibited a substantial angular dispersion, while the even number of phonon modes remained dispersion-less (Figure 7c).

### 4. Concluding Remarks

In summary, we exploited both phenomenological [19,54–68] and realistic [54] lattice dynamical models to report the results of systematic studies for comprehending the phonon traits of ideal and graded novel $(SiC)_m/(GeC)_n$ SLs. Unlike the basic features derived from M-LCM, there are many atypical phonon attributes of SLs (viz., the anisotropy of zone-center optical modes, acoustic-mode anti-crossing, mini-gap formation, confinement, and gap modes), which can only be ascertained by using realistic lattice dynamical models, considering appropriate symmetry and long-range Coulomb interactions [10,54,66–69]. Like many strained layer SLs (e.g., InN/GaN, AlN/GaN, etc.), the interfaces in SiC/GeC can act as potential scatters to play crucial roles in thermal energy transport. Previously, the In concentration gradient across $GaN/In_{0.43}Ga_{0.57}N/Al_{0.1}Ga_{0.9}N$ interfaces have been measured using HRTEM and Raman scattering spectroscopy, providing evidence of alloyed

interfacial regions [21]. Experimentally, the identification of interfacial phonon modes in SLs is complicated due to the zone-folding effect along the growth direction. Our theoretical study clearly shows that it is possible to modify the vibrational features of SLs in a controlled manner by appropriately adjusting m, n, and $\Delta$. In M-LCM, the impact of interfacial layer thickness $\Delta$ ($\equiv$1, 2, and 3 MLs) was considered here for simulating RIPs in the graded $(SiC)_{10-\Delta}/(Si_{0.5}Ge_{0.5}C)_{\Delta}/(GeC)_{10-\Delta}/(Si_{0.5}Ge_{0.5}C)_{\Delta}$ SLs by using a bond polarizability model. Virtual crystal approximation was adopted by assuming an equal proportion of SiC and GeC to describe the interfacial layer thickness, $\Delta$. Quite recently, Davydov et al. [10,69] have analyzed the Raman scattering spectra in $(GaN)_m/(AlN)_n$ SLs by using sharp interfaces and considering different degrees of interfacial diffusion. The authors of [10,69] have performed simulations of RIPs by using an ab initio approach, where the interfacial layer thickness is included in the framework of a random-element, iso-displacement model. The comparison of theoretical simulations with Raman scattering data on plasma-assisted MBE and MOVPE grown $(GaN)_m/(AlN)_n$ SLs [10,69] led the authors to conclude that the spectral region of COMs is quite sensitive to the degree of interface sharpness. Obviously, the results reported by Davydov et al. are in very good agreement with our simulations of the RIPs in both ideal and graded novel $(SiC)_m/(GeC)_n$ SLs. By varying $\Delta$, our calculations not only indicate the meaningful modifications in phonon dispersions $\omega_j^{SL}$ but also suggest that atomic displacements are responsible for causing the localization of COMs, which triggers significant enhancement in Raman scattering intensity features. The strong variations in RIPs can be used as a vital tool for Raman spectroscopists to perform similar measurements and test our theoretical conjectures for establishing interfacial structures in these and other SLs of technological importance. We hope that the contriving of phonon spectrum results in the novel $(SiC)_m/(GeC)_n$ SLs will create exciting opportunities among engineers and scientists to help improve the electrical and thermal properties of XC-based devices.

**Author Contributions:** Conceptualization, methodology, investigation, writing—original draft preparation, D.N.T.; conceptualization, methodology, writing—original draft preparation, writing—review and editing, supervision, P.B. All authors have read and agreed to the published version of the manuscript.

**Funding:** This research has received no external funding.

**Data Availability Statement:** The data that support the findings of this study are available from the corresponding author upon reasonable request.

**Acknowledgments:** D.N.T. wishes to thank Deanne Snavely, Dean College of Natural Science and Mathematics at Indiana University of Pennsylvania (IUP), for the travel support and the Innovation Grant that he received from the School of Graduate Studies, making this research possible. D.N.T. would also like to express his sincere gratitude to IUP undergraduate student, Sky Semonsky, for his help in plotting the atomic displacement graphs (Figure 3b) reported in this paper.

**Conflicts of Interest:** The authors declare no conflict of interest.

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
