# Peer review of "Evaluating Phonon Characteristics by Varying the Layer and Interfacial Thickness in Novel Carbon-Based Strained-Layer Superlattices"

_solids, doi:10.3390/solids4040018_

Round 1
Reviewer 1 Report
Previously published works by the authors:
-D. N. Talwar, J. Vac. Sci. Technol. A 40, 042704 (2022):
-D. N. Tawlar, P. Becla, Physica B 650, 414500 (2023);
What is new in the peer-reviewed article compared to previously published ones?
You also need to fix the affiliations.
What does "delta:" mean in Fig. 3a?
Probably, in Table 1, some of the columns are shifted one line down.
Author Response
Re: Our manuscript solids-2534149
Thank you very much for sending us Reviewer’s Comments that certainly helped us improve our manuscript. We have answered every question of the reviewers very carefully. All the changes in the revised version of the manuscript are made by using red color text.
Comments and Suggestions for Authors
Previously published works by the authors:
-D. N. Talwar, J. Vac. Sci. Technol. A 40, 042704 (2022):
-D. N. Tawlar, P. Becla, Physica B 650, 414500 (2023)
Authors response to Reviewer #1
Thank you very much for reading our manuscript. In the Reference section we have included the previously published work see Refs. [61-63].
Reviewer 2 Report
The manuscript solids-2534149 discusses on theoretical Raman spectra of (SiC)m/(GeC)n superlattices (SLs). Phonon properties including acoustic and optical phonon features are disclosed in this way by exploring (SiC)10-D/(Si0.5Ge0.5C)D/(GeC)10- D/(Si0.5Ge0.5C)D with changing D. a modified linear-chain model is applied in simulation.Phonon bands are displayed and Raman spectra with different superlattice parameters are shown.When the superlattice gets more layers, its Raman spectrum appears more peaks and peak shifting. SiC, GeC are calculated using a realistic rigid-ion model to show phonon dispersion and Debye temperature, (SiC)2/(GeC)2 SL phonon properties are also calculated. Calculating phonon properties of new SLs is interesting. This work is publishable after some minor improvement.
1. In Fig. 3. A)lable, D values are missing.
2. Authors should explain more clearly if there is a result identity between the modified linear-chain model and the realistic rigid-ion model.
3. Is it possible to do more comparison between theoretical results and other reported experimental results?
Author Response
Reviewer #2
Comments and Suggestions for Authors
The manuscript solids-2534149 discusses on theoretical Raman spectra of (SiC)m/(GeC)n superlattices (SLs). Phonon properties including acoustic and optical phonon features are disclosed in this way by exploring (SiC)10-D/(Si0.5Ge0.5C)D/(GeC)10- D/(Si0.5Ge0.5C)D with changing D. a modified linear-chain model is applied in simulation. Phonon bands are displayed and Raman spectra with different superlattice parameters are shown. When the superlattice gets more layers, its Raman spectrum appears more peaks and peak shifting. SiC, GeC are calculated using a realistic rigid-ion model to show phonon dispersion and Debye temperature, (SiC)2/(GeC)2 SL phonon properties are also calculated. Calculating phonon properties of new SLs is interesting. This work is publishable after some minor improvement.
- In Fig. 3. A)lable, Dvalues are missing.
- Authors should explain more clearly if there is a result identity between the modified linear-chain model and the realistic rigid-ion model.
- Is it possible to do more comparison between theoretical results and other reported experimental results?
Authors response to Reviewer #2
Thank you very much to your expert Reviewer #2 for reading our manuscript thoroughly and providing us excellent suggestions to improve our manuscript. We have answered every question carefully and modified the manuscript with appropriate changes (shown by red colored text) as suggested by Reviewer.
Thanks
- Yes, we have added new text and improved the manuscript significantly.
- The Fig. 3 a) now clearly shows the values of D used for simulating Raman intensity profiles for graded SLs.
- Yes, we have provided comparison of the M-LCM and RIM for phonon dispersions along the (001) direction (see page 11)
- Yes, now we have included the Experimental results of phonon dispersion of zb SiC Ref. [14] (see Fig. 4 a) while comparing our RIM calculations.
Reviewer 3 Report
The article presented for review is an interesting study, which in the future may be helpful in the interpretation of Raman spectra for structures based on silicon carbide and germanium. However, there are shortcomings that need to be corrected. The most important remark is that the title does not reflect the content of the manuscript. The manuscript did not assess the thickness of the interlayer, but (in its part) the influence of its different thickness on the shape of the Raman spectrum obtained in numerical analysis.
Detailed comments on the manuscript are provided below:
1) What is hidden under the subscripts "m" and "n" in the structure (SiC)m/(GeC)n? Please explain this in the abstract and in the introduction.
2) Has an attempt been made to confirm that the spectra presented (for example) in Figure 2 can be obtained for a real sample and at what wavelength of the scattered electromagnetic radiation?
3) The structure of the (SiC)10-D/(Si0.5Ge0.5C)D/(GeC)10-D/(Si0.5Ge0.5C)D layers is analyzed/discussed in the manuscript. What is the rationale for using the so-called interlayer at the end of this enumeration? From a practical point of view, no one will produce this layer.
4) Please check the references in the text of the manuscript (examples of errors/mistakes: line 136, 245).
5) Please correct/complete the descriptions in Figure 3.
6) Figure 5 - please explain exactly which data are the result of the authors' analyzes and which are taken from the literature and from which.
7) Please pay attention to the correct entering (writing) of units.
8) Please check the acknowledgments (I couldn't find figure 2d in the manuscript).
Author Response
Reviewer #3
Comments and Suggestions for Authors
The article presented for review is an interesting study, which in the future may be helpful in the interpretation of Raman spectra for structures based on silicon carbide and germanium. However, there are shortcomings that need to be corrected. The most important remark is that the title does not reflect the content of the manuscript. The manuscript did not assess the thickness of the interlayer, but (in its part) the influence of its different thickness on the shape of the Raman spectrum obtained in numerical analysis.
Detailed comments on the manuscript are provided below:
1) What is hidden under the subscripts "m" and "n" in the structure (SiC)m/(GeC)n? Please explain this in the abstract and in the introduction.
2) Has an attempt been made to confirm that the spectra presented (for example) in Figure 2 can be obtained for a real sample and at what wavelength of the scattered electromagnetic radiation?
3) The structure of the (SiC)10-D/(Si0.5Ge0.5C)D/(GeC)10-D/(Si0.5Ge0.5C)D layers is analyzed/discussed in the manuscript. What is the rationale for using the so-called interlayer at the end of this enumeration? From a practical point of view, no one will produce this layer.
4) Please check the references in the text of the manuscript (examples of errors/mistakes: line 136, 245).
5) Please correct/complete the descriptions in Figure 3.
6) Figure 5 - please explain exactly which data are the result of the authors' analyzes and which are taken from the literature and from which.
7) Please pay attention to the correct entering (writing) of units.
8) Please check the acknowledgments (I couldn't find figure 2d in the manuscript).
Authors response to Reviewer #3
Thank you very much to your expert Reviewer #3 for reading our manuscript thoroughly and providing us excellent suggestions to improve our manuscript. We have answered every question carefully and modified the manuscript with appropriate changes (shown by red colored text) as suggested by Reviewer.
Thanks
Yes, we have clearly stated the meaning of m, n in the abstract as well as in the revised version of the text.
- Thanks, yes we have clearly stated the meaning of m, n both in the abstract as well as in the revised version of the text.
- There are no Raman measurements reported in the novel C-based SiC/GeC SLs. However, our results are very well supported in recent articles by using Raman scattering measurements reported by Davydov et al. on GaN/AlN SLs (see: Refs. [10, 69]) where Raman intensities are simulated by using ab initio calculations. The interface layer GaAlN thickness was calculated using a random element iso-displacement model.
- The rationale to include thickness of interfacial layer D using virtual crystal approximation is reported in the revised version of the text. Similar ideas are also proposed in the recent works of Davydov et al. on GaN/AlN SLs (see: Refs. [10, 69]).
- Thanks, we have now very clearly checked all the Refs. reported in the revised version of our manuscript.
- Thanks, yes we have clearly checked Figs. 3 a) and 3 b) to make sure that the values of D are visible.
- Thanks, yes we have clearly checked Figs. 4-5 a) and 4-5 b) where the full lines (blue and red color) indicate our simulation compared with existing experimental data on zb SiC.
- Thanks, we have thoroughly checked all the units of various quantities reported in the revised text.
Thanks, we have made the correction of Fig. 3 b) in the acknowledgement section.
Round 2
Reviewer 1 Report
I have no more questions on this manucript.
Reviewer 3 Report
Thank you for your answers and corrections. Unfortunately, it is a pity that the authors do not have Raman spectra for the real SiC/GeC coating. Nevertheless, it accepts the manuscript as it is.